# Methylation of the Suppressor Gene *p16INK4a*: Mechanism and Consequences

**DOI:** 10.3390/biom10030446

**Published:** 2020-03-13

**Authors:** Alfonso Tramontano, Francesca Ludovica Boffo, Giusi Russo, Mariarosaria De Rosa, Ilaria Iodice, Antonio Pezone

**Affiliations:** 1Department of Precision Medicine University of Campania “L. Vanvitelli”, 80131 Naples, Italy; alfonsotramontano1989@gmail.com; 2Dipartimento di Medicina Molecolare e Biotecnologie Mediche, Istituto di Endocrinologia ed Oncologia Sperimentale del C.N.R., Università Federico II, 80131 Napoli, Italy; francescaludovicaboffo@gmail.com (F.L.B.); russo.giusi84@gmail.com (G.R.); meryderosa@gmail.com (M.D.R.); ilariaiodice@gmail.com (I.I.)

**Keywords:** DNA damage and repair, DNA methylation, gene expression

## Abstract

Tumor suppressor genes in the *CDKN2A/B* locus (*p15INK4b*, *p16INK4a*, and *p14ARF*) function as biological barriers to transformation and are the most frequently silenced or deleted genes in human cancers. This gene silencing frequently occurs due to DNA methylation of the promoter regions, although the underlying mechanism is currently unknown. We present evidence that methylation of *p16INK4a* promoter is associated with DNA damage caused by interference between transcription and replication processes. Inhibition of replication or transcription significantly reduces the DNA damage and CpGs methylation of the *p16INK4a* promoter. We conclude that de novo methylation of the promoter regions is dependent on local DNA damage. DNA methylation reduces the expression of *p16INK4a* and ultimately removes this barrier to oncogene-induced senescence.

## 1. Introduction

Cancer progression is characterized by the accumulation of genetic mutations and epigenetic alterations, which disrupt regulatory control mechanisms and profoundly affect cellular proliferation potential, growth signaling, neovascularization, and apoptosis [1,2,3].

In normal cells, progression into cell cycle is tightly regulated by a set of proteins that control the cell cycle checkpoints [4,5]. These checkpoints are often deregulated in cancer secondary to the silencing of various tumor suppressor genes [5,6]. The loss of tumor suppressor(s) through gene deletion, inactivating mutations, epigenetic silencing, or post-translational modification results in uncontrolled cellular growth. The progression of the mammalian cell cycle from G1 to DNA synthesis and eventually mitosis is regulated by cycling proteins and their catalytic subunits, referred to as cyclin-dependent kinases (CDKs) [7]. The genes encoding several CDK inhibitors CDKNs), such as *p15INK4b*, *p16INK4a*, and *p14ARF*, are frequently inactivated (silenced or deleted) in human cancers. The CDKNs encoded by these genes are bona fide tumor suppressors, and thus their inactivation is essential for tumor progression.

We have found that DNA damage and repair modify local DNA methylation profiles. In a precise time frame following DNA repair, transcription further remodels the methylation landscape, generating polymorphic methylation profiles that are transmitted to daughter cells. This association between DNA damage, repair, and methylation has been validated in several biological systems and cells [8,9,10].

Depending on the localization and density of the methylation sites in each cell clone, the expression of the repaired gene progressively changes [11,12,13,14]. Thus, methylated clones will dominate over others if silencing of the gene produces a strong positive selection.

The *CDKN2A/B* locus that encodes *p15INK4b*, *p16INK4a*, and *p14ARF* is frequently methylated in human cancers (resulting in gene silencing). Therefore, we wondered whether local DNA methylation at *p16INK4a* (the most frequently methylated gene encoding a CDKN) promoter regions in the *CDKN2A/B* locus is a consequence of local DNA damage and repair. DNA double strand breaks (DSBs) can originate from co-directional collisions between DNA and RNA polymerases [15], and interferences between these transcription and replication machines can be induced through the simultaneous induction of both these processes. For this reason, we searched the *CDKN2A/B* locus for origins of replication. The *CDKN2A/B* locus spans 25 Mb of genomic DNA on human chromosome 9 and includes two annotated replication origins in Hela cells, which are located 1 Kb upstream from the transcription start sites (TSSs) of *p15INK4b* and *p16INK4a/p14ARF* genes (Figure 1) [16].

To test our hypothesis, we induced transcription of these genes, by first blocking cell cycle progression and then rapidly inducing the re-entry. In order to accomplish this, first we determined the timing of *p15INK4b*, *p16INK4a*, and *p14ARF* gene induction during the cell cycle in both control and serum-starved Hela cells. Second, we induced progression into the cell cycle by adding serum to transiently starved cells, thereby inducing both the transcription of the *p15INK4b*, *p16INK4a*, and *p14ARF* genes and the onset of replication of DNA.

## 2. Materials and Methods

### 2.1. Cell Culture and Drugs Treatment

HeLa cells were cultured in DMEM medium with 4.5 g/L D-glucose and Pyruvate (Gibco, Carlsbad, CA, USA) complemented with 100 u/mL of penicillin and 100 μg/mL of streptomycin (Gibco, Carlsbad, CA, USA), 2 mM L-glutamine (Gibco, Carlsbad, CA, USA), in the presence or the absence of 10% of FBS (South America origin, Brazil, Invitrogen, Rockville, MD, USA). All cultures were maintained in 37 °C at 5% CO2 humidified atmosphere. All drugs treatments were administered to adherent cells and dissolved in complete or starvation medium. The aphidicolin (Sigma-Aldrich, St Louis, MO, USA) at a final concentration of 1 μg/mL and the α-amanitine (Sigma-Aldrich, St Louis, MO, USA) at a final concentration of 2.5 μM were added to cell culture medium. The etoposide (Sigma- Aldrich, St Louis, MO, USA) was added to cell culture medium for 30 min, before harvesting them in an appropriate volume in order to reach the final concentration of 25 μM.

### 2.2. RNA Extraction and Analysis

Total RNA were extracted using TRI-REAGENT^®^ (Sigma-Aldrich, St Louis, MO, USA) solution, according to the manufacturer’s instruction. The nucleic acid quality was tested using NanoDrop 2000 (Thermo Scientific, Wilmington, DE, USA) by measuring the absorbance ratio at 260/230 nm and 260/280 nm. One microgram of total mRNAs was reverse transcribed using SensiFAST^®^ cDNA Synthesis Kit (Bioline, London, UK) according to the manufacturer’s instruction, redissolved in 20 μL of nuclease free water (Qiagen, Hilden, Germany). All PCR real time experiments were performed three times on a 7500 Real Times PCR System (Applied Biosystems, Foster City, CA, USA) using the SYBR^®^ Green-detection system (Roche, Penzberg, Germany). The complete list of oligonucleotides is reported in Table 1.

### 2.3. Protein Extraction and Western Blot Analysis

Cells lysis was carried out using RIPA-Buffer (Sigma-Aldrich, St Louis, MO, USA) and the protein concentrations were determined by Bio-Rad protein assay. Equal amounts of denatured proteins were subjected to SDS PAGE 10% polyacrylamide gel. Proteins were visualized using ECL substrate (Euroclone, Milano, Italy) and ECL chemiluminescence film (Fujifilm^®^, Tokyo, Japan). Phospho S1981 ATM antibody (Abcam, Cambridge, UK, ab81292) and the antibody targeting total ATM (Abcam, Cambridge, UK, ab199726) were used to determine the fraction of active ATM. Both proteins were then normalized to the β- Actin levels (Sigma-Aldrich, St Louis, MO, USA, A1978) as loading control. Image J software was used to measure the relative density of immunoblot bands and to calculate their ratio by densitometry.

### 2.4. Chromatin Immunoprecipitation

HeLa cells were starved from FBS, as indicated in the legends of the figures. Cells were washed with PBS and fixed with formaldehyde at final concentration of 1%, then the reaction was quenched by adding 125 mM glycine.

Nuclei were isolated with a mild lysis buffer (Proteinase inhibitor cocktail, 10 mM Tris pH 8.0, 10 mM NaCl, NP40 0.2%, PMSF 1%) and then they were fragmented by sonication (Bioruptor^®^ Pico Sonicator, Diagenode, Ougrèe, Belgium). An aliquot for each sample was stored as INPUT, and the remaining part was immunoprecipitated using anti gamma H2A.X (phospho S139) antibody (Cell Signaling, Danvers, MA, USA, 2577). All samples were processed using a chromatin immunoprecipitation (ChIP) assay kit (Upstate Biotechnology, Lake Placid, NY, USA) according to the manufacturer’s protocol.

The results were obtained by Real Time PCR assay, as indicated in the text. The input DNA samples values were used to normalize the ChIP samples. The protein concentration of chromatin was comparable in all the samples. All values represent the average of at least three independent experiments.

### 2.5. Flow Cytometry Analysis

Cells were harvested and fixed with 70% cold ethanol and PBS solution in a ratio 9:1 at 4 °C overnight. After being washed in PBS, the cells were incubated in 500 μL of staining solution (20 μg/mL propidium iodide; 0.1–0.2 mg/mL RNaseA; TRITON 0.1%) at room temperature for 30 min. Then, the samples were analyzed with the FACSCAN (BD, Heidelberg, Germany) and the cell cycle distributions were determined with the software WINMDI.

### 2.6. MeDIP Assay

Cells were treated as indicated in the figure legends. A total of ~2.6 × 10^6^ cells were harvested and genomic DNA was extracted from cell pellets, purified by phenol/chloroform/isoamyl alcohol extraction end precipitated with ethanol. The resulting DNA was then re-dissolved in Tris-EDTA buffer (Sigma-Aldrich, St Louis, MO, USA). 7.7 micrograms of total genomic DNA was digested in 100 μL for 16 h at 37 °C with Restriction Endonuclease mix containing 30 U each of Eco RI, Hind III, Hpa II (Roche, Penzberg, Germany), phenol/chloroform extracted, ethanol precipitated and resuspended in 50 μL of TE buffer (10 mM Tris pH 8.0, 1 mM EDTA). An aliquot (1/10) of digested DNA was used as input control to determine the DNA concentration and the digestion efficiency. The remaining DNA was diluted in 500 μL of Immunoprecipitation buffer (0.15% SDS, 1% Triton X- 100, 150 mM NaCl, 1 mM EDTA pH 8.0, 0.5 mM EGTA pH 8.0, 10 mM Tris pH 8.0, 0.1% BSA, 7 mM NaOH) and incubated at 95 °C for 10 min before the immunoprecipitation with 5 µg of 5-methyl cytosine antibody (Abcam, Cambridge, UK, ab10805). As a control, 5 µg of normal mouse IgG (Santa Cruz, Dallas, TX, USA, sc-2025) was included. All samples were analyzed by Real Time PCR.

## 3. Statistical Analysis

All data are presented are mean ± standard deviation of three independent experiments in triplicate (n = 9). Statistical significance between groups was determined using Student’s t test (matched pairs test or unmatched test were used as indicated in figure legends).

## 4. Results

### 4.1. Serum Deprivation Upregulates p15INK4b, p16INK4a, and p14ARF or CDKN2A/B Gene Expression

Hela Cells were starved for 2 h and then exposed to serum for either 2 h or 4 h. Cytofluorimetric analysis showed that cells were in G0/G1 phase after 2 h of serum starvation, and the addition of serum for either 2 h or 4 h induced progression through the cell cycle (Figure 2A, compare G0/G1 - S and G2/M - S ratios). The observed changes to the cell cycle were modest, as approximately only 10–15% of the cells were synchronized followed by 2 h of serum starvation, however these differences were significant and reproducible as demonstrated by p values below 0.05 (Figure 2A). In these starved and then induced cells, we found a significant induction of transcription evidenced by the accumulation of mRNA encoded by *p15INK4b*, *p16INK4a*, and *p14ARF* genes (Figure 2B). Re-addition of serum forced a fraction of these cells to re-enter the cell cycle and dramatically reduced *p15INK4b*, *p16INK4a*, and *p14ARF* expression. Under the same conditions, the levels of mRNA encoding for p21Cip1 did not change (Figure 2B).

### 4.2. Rapid Reentry into the Cycle Induces DNA Damage

To investigate the consequences of switching on and off *CDKN2A/B* locus on and off in HeLa cells, we performed a chromatin immunoprecipitation (ChIP) assay with antibodies against phosphorylated γH2aX protein, which detects DSBs [17]. Specifically, we analyzed the promoter regions of the *p15INK4b*, *p16INK4a*, and *p14ARF* genes under several different starvation and growth conditions. Figure 2C shows that phosphorylated γH2AX selectively accumulated at the promoter region of *p15INK4b*, *p16INK4a* and *p14ARF* genes at the onset of S1 phase (Figure 2B). Under these same conditions, γH2AX was absent from the *p21Cip1* promoter region (Figure 2C). To obtain independent evidence demonstrating DNA damage following 2 h of starvation plus 2 h of serum (2 h + 2 h), we performed a western blot analysis using antibodies to phosphorylated ATM (pATM), which is the active form of ATM. Figure 2D shows a positive control with etoposide, which targets topoisomerase II, generating DBS and inducing ATM. The 2 h + 2 h conditions induced ATM phosphorylation, confirming DNA damage at the *p15INK4b*, *p16INK4a*, and *p14ARF* promoters through accumulation of phospho-γH2AX (Figure 2C). DNA damage was induced by rapid re-entry into the cell cycle (via 2 h + 2 h) and was not linearly dependent on the transcription efficiency or accumulation of mRNA corresponding to p15INK4b, p16INK4a, and p14ARF. Extensive DNA damage was observed to occur at the *p14ARF*, *p16INK4a*, and *p15INK4b* promoters, but was not evident at the *p21Cip1* promoter (Figure 2C).

### 4.3. Inhibition of DNA Polymerase or RNA Polymerase Reduces γH2AX Accumulation at p16INK4a Promoter

The data shown above suggest that, at the onset of S phase, interference of DNA polymerase by RNA polymerase, which was still engaged at the *p16INK4a* promoters, might be the cause of DNA damage induced by short cycles of starvation and serum stimulation (2 h + 2 h). If this is the case, inhibition of either DNA or RNA polymerases should significantly reduce the accumulation of γH2AX at these regions. To investigate this hypothesis, we performed ChIP assays with γH2AX antibodies interrogating *p16INK4a* promoter using two polymerases inhibitors, α-amanitin (an inhibitor of RNA polymerase [18]), and aphidicolin (a DNA polymerase inhibitor [19]). We chose to analyze the *p16INK4a* promoter, and not *p14ARF/p15INK4b* promoter. This choice was made because the *p14ARF* promoter area contains two TSSs that direct the synthesis of sense (*p14ARF*) and antisense (*CDKN2B-AS1/ANRIL*) RNAs, the latter of which can regulate *p15INK4b* expression, but not *p16INK4a* or *p14ARF* expression, under various conditions in ATM-dependent manner through the recruitment of PRC2 on the *p15INK4b* promoter [20]. The presence of two transcription initiation complexes on the *p14ARF* promoter, as well as transcription of *CDKN2B-AS1/ANRIL*, would have potentially complicated the interpretation of results. Figure 3A shows that, during serum starvation, there was a significant increase in γH2AX accumulation at the *p16INK4a* promoter in cells exposed to aphidicolin while, in cells exposed to conditions of maximal γH2AX accumulation (2 h + 2 h), both aphidicolin and α-amanitin significantly reduced the accumulation of γH2AX at the same site. Note that exposure to 1 μg/2 h α-amanitin did not inhibit the basal transcriptional level; only the transcription induced by progression in the cell cycle was inhibited. It is worth noting that aphidicolin inhibits DNA synthesis during repair, but does not induce damage in the absence of other factors per se, such as oxidative stress [21], thus the induction of DNA damage by aphidicolin in G0 cells was caused by the inhibition of DNA repair [20].

Following 2 h of serum starvation, treatment with aphidicolin induced p16INK4a mRNA levels secondary to checkpoint activation in G0/G1 cells by unrepaired DNA damage [22]. Conversely, α-amanitin inhibited the accumulation of p16INK4 mRNA following starvation as expected (Figure 3B).

### 4.4. Methylation Associated with DNA Damage at p16INK4a Promoter

The data shown above demonstrate that rapid cycles of starvation and serum re-addition induced DNA damage at the *p16INK4a* promoter (Figure 3C), while the inhibition of DNA or RNA polymerases significantly reduced this damage (Figure 3A). To test whether this DNA damage at the *p16INK4a* promoter was associated with methylation at the same genomic region, we measured the degree of DNA methylation in cells exposed to the 2 h + 2 h conditions through methylated DNA immunoprecipitation (MeDIP) using an anti-5-methylcytosine antibody to immunoprecipitate the chromatin of the *p16INK4a* promoter. Figure 3C shows that: (1) aphidicolin and α-amanitin increased the signal of the *p16INK4a* promoter under basal conditions; (2) starvation reduced the signal induced by α-amanitin; (3) in cells exposed to consecutively starvation and re-addition of serum (2 h + 2 h), both α-amanitin and aphidicolin substantially reduced the MEDIP signal (Figure 3C) and (4) α-amanitin and aphidicolin increased the MEDIP signal in basal and starved cells, respectively (Figure 3C). We conclude that the observed DNA damage was associated with an increase in the MEDIP signal, which implies an increase in the methylation of the DNA. Cells subjected to rapid cycles of starvation and serum stimulation accumulated high levels of γH2aX and CpG methylation (Figure 2C and Figure 3C).

## 5. Discussion

### 5.1. Molecular Collisions as Sources of DNA Damage

Every day, the DNA of our cells is subjected to a wide array of potentially damaging effects [23] induced by both exogenous and endogenous sources. The continuous exposure of cells to DNA damaging agents represents the most likely cause of degenerative diseases and aging.

Many types of lesions can affect DNA, however DSBs are the most dangerous events altering genome stability. Previous studies have described molecular collisions between the DNA replication apparatus and the RNA polymerase transcriptional complex as a potential cause of DSBs in bacteria [22,24]. Two types of collisions have been described, namely co-directional and head-on collisions. Differences in the rates of polymerization by DNA (1K nucleotides/sec) and RNA (80 nucleotides/sec) polymerases in bacteria [25] could justify the existence of co-directional collision when the enzymes work together on the leading strand (Figure 4). If the DNA segments are highly transcribed [26], they can become a hotspot for co-directional conflicts. Additionally, when a gene is encoded by the lagging strand head-on orientation collisions occur more frequently (Figure 4) [27], thus slowing down replication fork progression [28]. In mammalian cells, the replication fork pauses (RFP) caused by head-on collision have been associated with increased levels of homologous recombination (HR) at these site [28]. DNA regions with high levels of RFP activity coupled with DNA break formation are known as common fragile sites (CFS) [29].

### 5.2. DNA Damage and DNA Methylation

Molecular collisions can cause DNA damage and aberrant DNA methylation, both of which are important in the promotion of tumor evolution and chemoresistance. The data shown here demonstrate that the methylation of *p16INK4a* in DNA segments close to the TSS is associated with high rate of DNA damage.

DNA DSBs may result from collisions between transcription and replication enzymes. DBS at promoter sites are frequently repaired by HR [28,29,30,31]. Our previous data show that DNA damage repaired by HR can lead to the generation of methylated and non-methylated molecules that segregate in low and high expressor clones. Remodeling of the methylation landscape through transcription further increases the polymorphism of somatic DNA methylation [11,12,13,14]. HR-directed DNA repair generates cells that possess polymorphic epi-alleles, differing from each other only in their methylation, and thus each cell expresses a discrete level of the repaired gene. Somatic methylated epi-alleles are usually stably inherited and, in the absence of selection, become randomly distributed in cell populations [13,14,15]. However, if the epi-allele confers a growth advantage, for example by reducing the expression of a suppressor gene, the frequency of the epi-alleles will increase overtime secondary to positive selection.

In conclusion, we believe that the data shown here describe a frequent mechanism of de novo methylation of the *p16INK4* suppressor gene. The methylation trait detailed here leads to powerful positive selection of the cell clones in normal or tumor cells, suggesting existence of genomic sites which are more prone to methylation than other regions dependent upon their distance from a TSS and a replication origin.

## Figures and Tables

**Figure 1 biomolecules-10-00446-f001:**
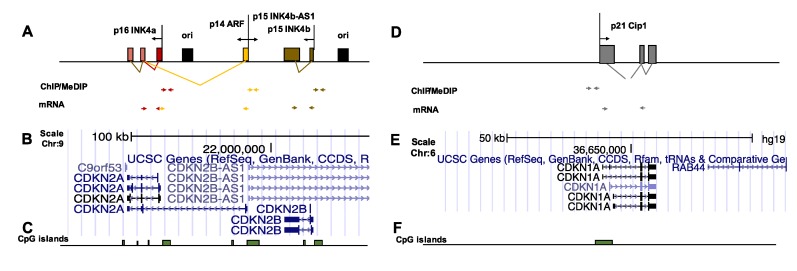
(**A**,**D**) Genomic structure of the locus of *CDKN2A/B* on chromosome 9 and *CDKN1A* (*p21CIP1*) on chromosome 6. The arrows show the location of the primers used for RNA, chromatin immunoprecipitation (ChIP), and MEDIP analysis. (**B**,**E**) show the map and the structure of RNA transcripts in both loci derived from UCSC Genome Browser. (**C**,**F**) show the CpG islands in both loci derived from UCSC Genome Browser.

**Figure 2 biomolecules-10-00446-f002:**
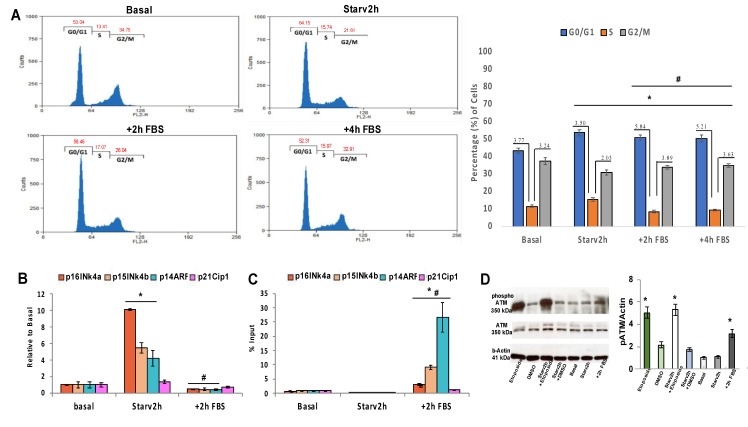
Serum addition to transiently starved cells induced DNA synthesis, transcription and DNA damage in *CDKN2A/B* genes. (**A**) Total RNA was extracted and analyzed by qRT-PCR with primers corresponding to *p16INK4a* (*CDKN2A*), *p15INK4b* (*CDKN2B*), *p14ARF* (*CDKN2A-ARF)*, and *p21Cip1* (*CDKN1A*) (Table 1). (**B**) Cell cycle was analyzed with the FACSCAN (BD, Heidelberg, Germany) and the software WINMDI. (**C**) Chromatin immunoprecipitation with phospho-γH2AX antibody was performed as described in the Materials and Methods. The input DNA values were used to normalize the ChIP samples. (**D**) Immunoblot analysis of phospho S1981 ATM and total ATM protein levels in HeLa cell extracts, derived from cells treated with DMSO, or starved or exposed to serum as indicated in the Materials and Methods and in the text. β-Actin was used as loading control. The statistical analysis was performed on three independent experiments in triplicate (n = 9; mean ± SD); * *p* < 0.05 (matched pairs *t*-test) compared to Basal; # *p* < 0.05 (matched pairs *t*-test) compared to Starv 2 h.

**Figure 3 biomolecules-10-00446-f003:**
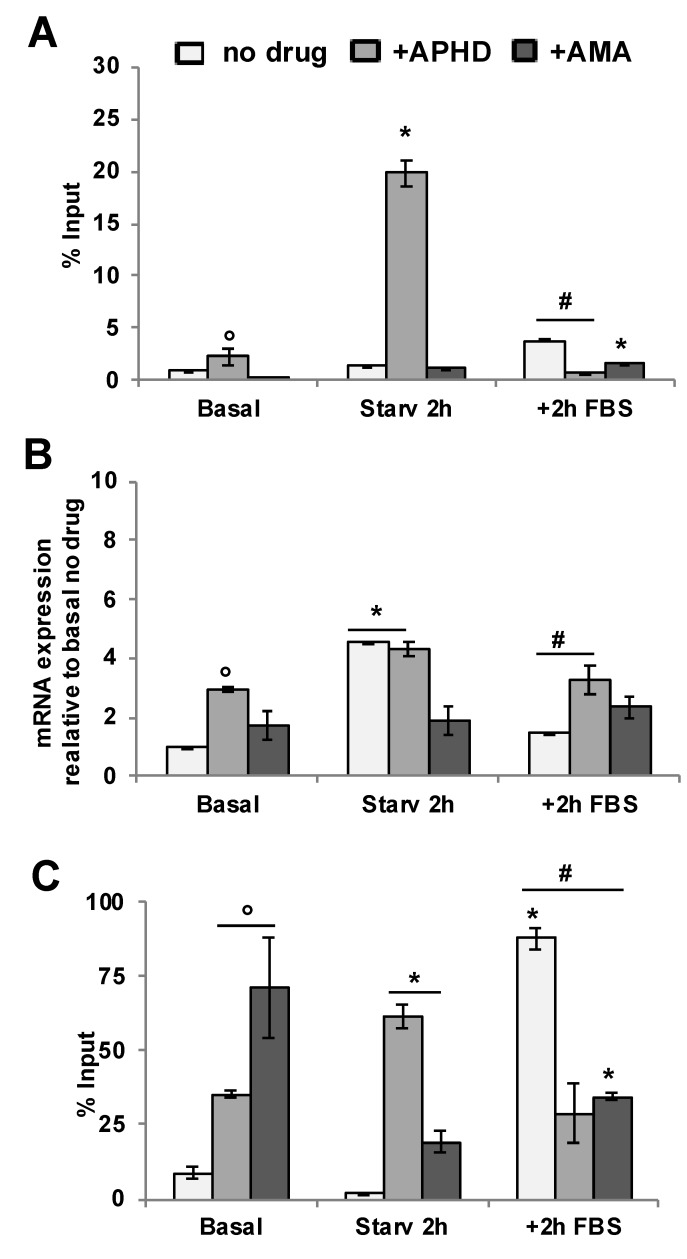
Inhibition of transcription and replication induced DNA damage at *p16INK4a* promoter. Hela cells were serum starved for 2 h before stimulated with 10% FBS for 2 h. (**A**) Chromatin immunoprecipitation with γH2AX antibody was performed as described in Materials and Methods. (**B**) Total RNA was extracted and analyzed by qRT-PCR with p16INK4a primers (Figure 1 and Table 1). (**C**) Medip Assay of Hela cell was performed as described in Materials and Methods. Genomic DNA was incubated at 95 °C for 10 min before the immunoprecipitation with 5 µg of anti 5-methyl cytosine antibody. The statistical analysis derives from three independent experiments in triplicate (n = 9; mean ± SD); ° *p* < 0.05 (matched pairs t-test) compared to “no drug” Basal; * *p* < 0.05 (matched pairs *t*-test) compared to each Basal; # *p* < 0.05 (matched pairs *t*-test) compared to Starv 2 h.

**Figure 4 biomolecules-10-00446-f004:**
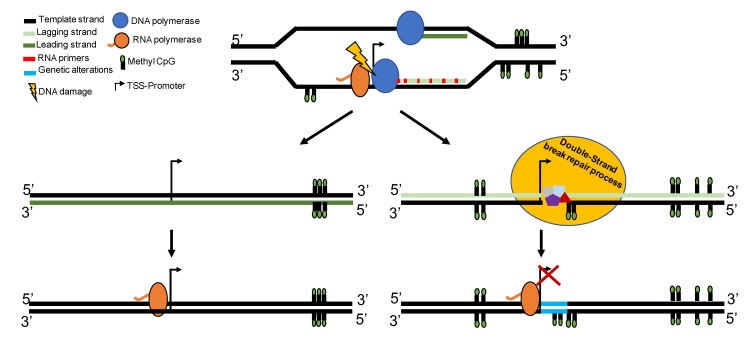
Graphical abstract. DNA damage caused by DNA and RNA polymerases collisions and DNA repair (Figure 2) are associated with de novo methylation of the repaired DNA segment (Figure 3). Inhibition of replication or transcription significantly reduces the damage and CpG methylation of the region (Figure 3). De novo methylation of the promoter regions reduces the expression of the suppressor genes, and ultimately removes the barrier to oncogene-induced senescence.

**Table 1 biomolecules-10-00446-t001:** Complete list of DNA oligonucleotides. On the left is shown the primer identification tag (ID) containing the name of gene; on the right, the corresponding DNA sequence.

	**Primers for mRNA**
18s fw	5′-GCG CTA CAC TGA CTG GCT C-3′
18s rv	5′-CAT CCA ATC GGT AGT AGC GAC-3′
p16INK4a fw	5′-TGG AGG CGG GGG CGC TGC CCA-3′
p16INK4a rv	5′-TCG TGC ACG GGT CGG GTG AGA-3′
p15INK4b fw	5′-AGT GGA GAA GGT GCG ACA GCT-3′
p15INK4b rv	5′-TCG GGT GAG AGT GGC AGG GT-3′
p14ARF fw	5′-CTC GTG CTG ATG CTA CTG AG-3′
p14ARF rv	5′-TCG TGC ACG GGT CGG GTG AGA-3′
p21Cip1 fw	5′-GAC AGC AGA GGA AGA CCA TG-3′
p21Cip1 rv	5′-CTC TTG GAG AAG ATC AGC CGG C-3′
	**Primers for ChIP**
p16INK4a fw	5′-GCC ATA CTT TCC CTA TGA CAC-3′
p16INK4a rv	5′-GAG CCA GCG TTG GCA AGG AAG-3′
p15INK4b fw	5′-GCG GGG ACT AGT GGA GAA G-3′
p15INK4b rv	5′-CTC CCG AAA CGG TTG ACT C-3′
p14ARF fw	5′-CCA GAA AGG ATC GGT GAT GT-3′
p14ARF rv	5′-ACG TTC TCT CTC CGG TCT CC-3′
p21Cip1 fw	5′-ATG TGT CCA GCG CAC CAA CG-3′
p21Cip1 rv	5′-AGC TCA GCG CGG CCC TGA TAT AC-3′

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
