# Peer review of "Methylation of the Suppressor Gene p16INK4a: Mechanism and Consequences"

_biomolecules, 2020, doi:10.3390/biom10030446_

Round 1

Reviewer 1 Report

Tramontano et al present a study to demonstrate how DNA and RNA polymerase collisions can lead to DNA double strand breaks and eventually result in de novo methylation, in the promoter regions of important tumor suppressor genes. The work is interesting and scientifically sound but still, I think that several changes should be made so that it can be published:

ABSTRACT: There is an erratum in the fourth line. Additionally, it is not very clear which genes/proteins have been analyzed and there is no summary of the methods used. Thus, the abstract is not as informative as it should.

INTRODUCTION: Again, the information about the genes/proteins is a bit confusing. I would specify which genes encode which proteins, which are being tested as test or candidate genes and which as controls… Also, I miss a hypothesis at the end of the introduction. It seems to me that the introduction finishes abruptly.

METHODS: I would specify in the numerous parts through the Methods section in which the authors state “…as indicated in the legends/text”. You should provide this information totally or omit. On the other hand, the authors mention that they performed several of the experiments in “at least three independent experiments”. What does this mean exactly? Could you specify? How did you decide the number of replicates? Finally for this section, I would include the reference number of each Ab in each corresponding subsection in Methods.

RESULTS AND DISCUSSION: The results look good. However, I miss more literature so that non-expert readership can understand the novelty/originality and impact of the findings. Have DNA damage and methylation been related before elsewhere? In which context/experimental design? And in which genes? Same about DNA/RNA polymerase collisions and DNA damage novelty: there is insufficient literature for non-specialized readers (I would add a few lines in the discussion or in the introduction).

Author Response

Point 1: ABSTRACT: There is an erratum in the fourth line.

Response 1: corrected

Point 2: Additionally, it is not very clear which genes/proteins have been analyzed and there is no summary of the methods used.

Response 2: We have added the new Fig.1 , which describes the structure of the locus,  the location of the transcription start sites, the sense and antisense encoded RNAs and the location of annotated replication origins  in Hela cells (refs 4 and 16). We have also uniformed the labelling of the genes throughout  manuscript as CDKN2A/B locus and p15INK4B, p16INK4A and p14ARF. Furthermore, a summary of methods is included  in the introduction  (line 92).

Point 3: INTRODUCTION: Again, the information about the genes/proteins is a bit confusing. I would specify which genes encode which proteins, which are being tested as test or candidate genes and which as controls… Also, I miss a hypothesis at the end of the introduction. It seems to me that the introduction finishes abruptly

Response 3: We have modified the labeling of genes using CDKN2A/B locus (p15INK4B, p16INK4A and p14ARF). In the introduction we have specified the working hypothesis (from line 92).

Point 4: METHODS: I would specify in the numerous parts through the Methods section in which the authors state “…as indicated in the legends/text”. You should provide this information totally or omit. On the other hand, the authors mention that they performed several of the experiments in “at least three independent experiments”. What does this mean exactly? Could you specify? How did you decide the number of replicates? Finally for this section, I would include the reference number of each Ab in each corresponding subsection in Methods.

Response 4: All data are presented are mean ± standard deviation in at least three experiments in triplicate (n≥9). We have indicated the reference number of each Ab in methods.

Point 5: RESULTS AND DISCUSSION: The results look good. However, I miss more literature so that non-expert readership can understand the novelty/originality and impact of the findings. Have DNA damage and methylation been related before elsewhere? In which context/experimental design? And in which genes? Same about DNA/RNA polymerase collisions and DNA damage novelty: there is insufficient literature for non-specialized readers (I would add a few lines in the discussion or in the introduction).

Response 5: The relation between DNA damage and methylation has been confirmed in several systems and independent observations (refs 8, 9, 10, 11, 12, 13, 14). We have quoted  a reference in which the collisions were demonstrated in E.coli (refs 26, 31)

Reviewer 2 Report

Comments on “Methylation of CDKN2A-B Suppressor Genes: Mechanism and Consequences”

In the manuscript by Tramontano et al., the authors study a potential mechanism by which silencing of the genes encoding the tumor suppressors CDKN2A/2B and ARF might occur in human cancer cells. The authors try to demonstrate at least parts of a cascade of events that built on one another, i.e. collision of the DNA replication and DNA transcription machinery at specific loci, formation of DNA double strand breaks (DSBs), repair of the damaged site, hypermethylation of the repaired region, and, as a result transcriptional silencing of the methylated locus. Indeed, the authors showed in their preceding studies that DNA DSB repair triggers stable hypermethylation of a repaired region (e.g. PMID: 27629060).

The mechanistic cascade of aberrant silencing of tumor suppressor genes on itself as well as its potential implications in tumor progression is of certain relevance. However, the authors do not show convincing data supporting their hypothesis.

Main issues:

The authors serum starve HeLa cells to synchronize the cells in G0/G1 to induce transcription of CDKN2A/2B and CDKN2A-ARF followed by addition of serum to induce S-phase progression, i.e. DNA replication. This set up should boost collisions of the two polymerase machineries and induce DSBs. While the expression of the tested genes is indeed induced by serum starvation (Figure 1B), I cannot see any induction of DNA replication by the addition of serum. The cell cycle profiles in Figure 1A do not show obvious differences. Moreover, the statistics of the cell cycle quantifications in the same panel are misleading since the differences are obviously marginal, especially for S-phase progression. How can the authors conclude that the DNA damage as well as the transcriptional silencing they observe 2h post serum addition (Figure 1B-C) is due to colliding replication and transcription machineries and not due to any other (indirect) effect by serum starvation and/or addition? This referee issue challenges the central hypothesis of the authors and demands convincing experimental evidence. The authors focus in Figure 2 on results they obtained from the CDKN2A locus only. Why? Although by serum starvation expression of this gene had the highest induction of all four genes tested, DSBs were appr. 2-fold and 5-fold less induced as compared to the CDKN2B and CDKN2A-ARF regions, respectively. The choice of a rather less damaged region to demonstrate the key point of the paper, i.e. dependence of the damage accumulation on the replication and/or transcription machinery, seems to me illogic. Thus, the authors should show in Figure 2 the analysis with all the three damaged regions they how in Figure 1C. Treatment of cells with aphidicolin and α-amanitin induced huge and unwanted side effects towards DNA damage accumulation (aphidicolin, α-amanitin, Figure 2A), expression of CDKN2A (aphidicolin, Figure 2B) and methylation of the CDKN2A locus (aphidicolin, α-amanitin, Figure 2C) in control conditions. In addition, α-amanitin seems not to block transcription of CDKN2A in basal conditions but rather boosts its expression by 2-fold (Figure 2B). How can the authors be sure that any effect after 2h of serum starvation/addition is specifically due to inhibition of replication and/or transcription and not to the side effects of the drugs? The ChIP (and presumably MeDIP) primers for CDKN2A align in the middle of intron 1 of CDKN2A but not in the promoter region of this gene as stated by the authors (e.g. lines 165-166, 209 and 217-218). What promoter region are the authors referring to? The title of the manuscript “Methylation of CDKN2A-B Suppressor Genes…” does not match to the content of the study since methylation changes are shown solely for a small region of CDKN2A (intronic region?).

Minor issues:

The labeling of most panels misses information. “Percent” (Figure 1A) is suboptimal to indicate percentage of cells in the respective cell cycles; “Relative to Basal” (Figure 1B) is not sufficient to specify relative mRNA levels of the tested genes to basal conditions set to 1; “% Input” is not enough to describe a gH2AX ChIP or a MeDIP result. (Figure 1C, 2A and C)? Why do the authors abbreviate aphidicolin as APDH but not APHD? Any details on duration of the drug treatments are missing in the methods section. The number of asteriks in figures is usually used to distinguish different classes of significances (e.g. *<0.05, **<0.001 etc.) but not to point on different pairings. Moreover, also the usage of three stars as in Figure 2C is confusing. The legend and main text to Figure 1 is mixed up (e.g. lines 148-151, 166 and 170). The legend to Figure 2 (see lines 196 and 199) describes experiments for gene loci that are actually not shown in the figure. There are formatting errors in the pdf-file (e.g. lines 61, 66, 67, 70).

Author Response

Response to Reviewer 2 Comments

Point 1: The authors serum starve HeLa cells to synchronize the cells in G0/G1 to induce transcription of CDKN2A/2B and CDKN2A-ARF followed by addition of serum to induce S-phase progression, i.e. DNA replication. This set up should boost collisions of the two polymerase machineries and induce DSBs. While the expression of the tested genes is indeed induced by serum starvation (Figure 1B), 1. I cannot see any induction of DNA replication by the addition of serum. The cell cycle profiles in Figure 1A do not show obvious differences. Moreover, the statistics of the cell cycle quantifications in the same panel are misleading since the differences are obviously marginal, especially for S-phase progression. How can the authors conclude that the DNA damage as well as the transcriptional silencing they observe 2h post serum addition (Figure 1B-C) is due to colliding replication and transcription machineries and not due to any other (indirect) effect by serum starvation and/or addition? This referee issue challenges the central hypothesis of the authors and demands convincing experimental evidence.

Response 1: The induction of cell cycle progression is documented by the changes of  the ratio G1/S (increase upon short starvation and reduction of G2/M) and S/G2-M (increase upon serum addition). The variations are small but reproducible in technical and biological replicates. The small fraction of progressing cells is essentially due to the short time (2h)  serum re-addition. We  have chosen such a tight and early window to detect the earliest replication timing involving early origins. Under these conditions we are observing approx.  10% of the cells. WE have added the ratio G0/G1 – S and G2/M – S in the figure.

Point 2: The authors focus in Figure 2 on results they obtained from the CDKN2A locus only. Why? Although by serum starvation expression of this gene had the highest induction of all four genes tested, DSBs were appr. 2-fold and 5-fold less induced as compared to the CDKN2B and CDKN2A-ARF regions, respectively. The choice of a rather less damaged region to demonstrate the key point of the paper, i.e. dependence of the damage accumulation on the replication and/or transcription machinery, seems to me illogic. Thus, the authors should show in Figure 2 the analysis with all the three damaged regions they how in Figure 1C.

Response 2: We have explained in the text (line 84) the reason why we have specifically analyzed, p16INK4A and not p14ARF. p14 promoter contains two TSS, which work on both strands in opposite directions (Fig.1). The robust damage accumulated at this p14 site may be the result of this bivalent TSS and not a simple mechanism we would like to test. p16 is less transcribed or damaged compared to p14 , but is more like average transcribed genes with a single TSS and it is the most silenced gene in cancers (doi:10.1200/JCO.1998.16.3.1197, doi:10.1006/excr.2000.5149).

We plan eventually , to systematically analyze TS sites  close to replication origins  in the genome and assess DNA damage and methylation after serial pulses of serum and starvation. We also added in the text that TSS in mammalian genomes are overlapping with replication origins and represent a physical border between hypomethylated and hypermethylated DNA domains (refs 4 and 16).

Point 3: Treatment of cells with aphidicolin and α-amanitin induced huge and unwanted side effects towards DNA damage accumulation (aphidicolin, α-amanitin, Figure 2A), expression of CDKN2A (aphidicolin, Figure 2B) and methylation of the CDKN2A locus (aphidicolin, α-amanitin, Figure 2C) in control conditions. In addition, α-amanitin seems not to block transcription of CDKN2A in basal conditions but rather boosts its expression by 2-fold (Figure 2B). How can the authors be sure that any effect after 2h of serum starvation/addition is specifically due to inhibition of replication and/or transcription and not to the side effects of the drugs?

Response 3: We have tested the drugs in several systems (induced by serum or by nuclear hormones) and we find that the amanitin blocks the induced transcription while the basal in some cases can also increase due to synchronization of RNA polymerase already engaged in active transcription, which is a a-amanitin resistant.

Point 4: The ChIP (and presumably MeDIP) primers for CDKN2A align in the middle of intron 1 of CDKN2A but not in the promoter region of this gene as stated by the authors (e.g. lines 165-166, 209 and 217-218). What promoter region are the authors referring to?

Response 4: We have added new Fig.1 which illustrates the localization of the primers. The intron  in p14 contains the p16 TATAA and TSS of p16.

Point 5: The title of the manuscript “Methylation of CDKN2A-B Suppressor Genes…” does not match to the content of the study since methylation changes are shown solely for a small region of CDKN2A(intronic region?).

Response 5: WE have changed the title and specified  that we have analysed the segment surrounding the TSS of  p16  

Point 6: The labeling of most panels misses information. “Percent” (Figure 1A) is suboptimal to indicate percentage of cells in the respective cell cycles;

Response 6: We have changed the labeling “Percent“ with “Percentage (%) of Cells”

Point 7: “Relative to Basal” (Figure 1B) is not sufficient to specify relative mRNA levels of the tested genes to basal conditions set to 1;

Response 7: Relative basal was used to normalize for different primers with different Tm.

Point 8: “% Input” is not enough to describe a gH2AX ChIP or a MeDIP result. (Figure 1C, 2A and C)?

Response 8: % input is the usual normalization of CHIP reaction. H3 normalization may be not accurate because nucleosome may be unstable at the promoter region.

Point 9: Why do the authors abbreviate aphidicolin as APDH but not APHD? Any details on duration of the drug treatments are missing in the methods section. The number of asteriks in figures is usually used to distinguish different classes of significances (e.g. *<0.05, **<0.001 etc.) but not to point on different pairings.

Response 9: Done

Point 10: Moreover, also the usage of three stars as in Figure 2C is confusing. The legend and main text to Figure 1 is mixed up (e.g. lines 148-151, 166 and 170).

Response 10: Corrected

Point 11: The legend to Figure 2 (see lines 196 and 199) describes experiments for gene loci that are actually not shown in the figure.

Response 11: Corrected

Point 12: There are formatting errors in the pdf-file (e.g. lines 61, 66, 67, 70).

Response 12: Corrected

Round 2

Reviewer 1 Report

The manuscript has considerably improved after the thorough revision performed by the authors. Regarding the issue of the number of replicates, I think that stating that "at least 3 biological replicates were performed in technical triplicate" may not be precise enough. One could wonder how many experiments is observing at each individual plot and whether any "outlier" was removed, and facts like these could impact the significance of the results. In this context, I keep advising the authors to be as precise as they can.

Author Response

Point 1: Regarding the issue of the number of replicates, I think that stating that "at least 3 biological replicates were performed in technical triplicate" may not be precise enough. One could wonder how many experiments is observing at each individual plot and whether any "outlier" was removed, and facts like these could impact the significance of the results. In this context, I keep advising the authors to be as precise as they can.

Response 1: All data are presented are mean ± standard deviation of three independent experiments in triplicate (n=9). "Outliers" have not been removed, (see the SDs in Figure 2C and Figure 3C). The description of the analyzed samples has been corrected in the text.

Reviewer 2 Report

The authors could clarify most of my concerns. Thank you! However, I have noted a few flaws in the revised version that must be corrected before publication:

  1. Why are almost all primer sequences changed in the revised version? Where they all incorrect in the original manuscript? This is hard to believe.
  2. Why do the authors still not test/show the p15INK4b region in their drug treatment analysis (shown for p16INK4a in Figure 3)?
  3. Which primers did the authors use for the MeDIP analysis shown in Figure 3C?
  4. English grammar and typos should be corrected extensively throughout the text, e.g. from the abstract: “, and to date is not known the mechanism CDKN 2A-2B)”.

Author Response

Point 1.           Why are almost all primer sequences changed in the revised version? Where they all incorrect in the original manuscript? This is hard to believe.

Response 1: There was an error in the preparation of the table of the primers, which has been corrected. We apologize for not mentioning it in the rebuttal letter.

Point 2.           Why do the authors still not test/show the p15INK4b region in their drug treatment analysis (shown for p16INK4a in Figure 3)?

Response 2: We have stated in the text (pag. lines 318-325) the reason why we have specifically analyzed, p16INK4Aand not p14ARF/p15INK4b. p14ARF promoter area contains two TSSs that direct the synthesis of sense and antisense RNAs, p14ARF and CDKN2B-AS1/ANRIL, respectively,; CDKN2B-AS1/ANRIL regulates p15INK4b expression,  but not p16INK4A or p14ARF, in ATM-dependent manner through the recruitment of PRC2 at the p15INK4b promoter (10.1038/onc.2010.568. Epub 2010 Dec 13), 10.1016/j.cellsig.2013.02.006, 10.1007/s10238-012-0181-x). p16INK4Agene is transcribed from a single TSS and it is the most silenced gene in cancers (doi:10.1200/JCO.1998.16.3.1197, doi:10.1006/excr.2000.5149). We plan to systematically analyze TS sites  close to replication origins  in the genome and assess DNA damage and methylation after serial pulses of serum and starvation.

Point 3.           Which primers did the authors use for the MeDIP analysis shown in Figure 3C?

Response 3: The primers for MeDIP are those used for the ChIP. In the legend of Fig.1 it was indicated: “The arrows show the position of the primers used for the analysis of RNA, ChIP and MEDIP”, it was not shown in Figure 1. We have modified Figure 1 by adding the "MeDIP" label and the positions of the CpG islands with respect to the TSS.

Point 4.           English grammar and typos should be corrected extensively throughout the text, e.g. from the abstract: “, and to date is not known the mechanism CDKN 2A-2B)”.

Response 4: The text has been corrected by professional editors (we can include the statement).
